# Patterns of Elder Caregiving Among Nigerians: An Integrative Review

**DOI:** 10.3390/ijerph23010002

**Published:** 2025-12-19

**Authors:** Chibuzo Stephanie Okigbo, Shannon Freeman, Dawn Hemingway, Jacqueline Holler, Glen Schmidt

**Affiliations:** 1Department of Health Sciences, University of Northern British Columbia, 3333 University Way, Prince George, BC V2N 4Z9, Canada; 2School of Nursing, University of Northern British Columbia, 3333 University Way, Prince George, BC V2N 4Z9, Canada; shannon.freeman@unbc.ca; 3Department of Social Work, University of Northern British Columbia, 3333 University Way, Prince George, BC V2N 4Z9, Canada; dawn.hemingway@unbc.ca (D.H.); schmidt@unbc.ca (G.S.); 4Department of History, Women’s Studies and Gender Studies, University of Northern British Columbia, 3333 University Way, Prince George, BC V2N 4Z9, Canada; jacqueline.holler@unbc.ca

**Keywords:** elder care, Nigeria, caregiving patterns, caregiving, seniors

## Abstract

**Highlights:**

**Public health relevance—How does this work relate to a public health issue?**
Synthesizes evidence on elder caregiving within Nigeria’s rapidly aging population, highlighting how family-based care functions as a primary public health response in the absence of robust formal systems.Examines how cultural norms, migration, gender roles, and socioeconomic constraints shape care arrangements, access to services, and health outcomes for older adults in a low-resource context.

**Public health significance—Why is this work of significance to public health?**
Provides the first integrative, intersectional synthesis of elder caregiving in Nigeria, clarifying how cultural, familial, economic, psychosocial, and policy factors jointly determine health vulnerabilities and care inequities.Demonstrates how intersecting gendered, class-based, and regional disparities in caregiving contribute to unmet health and social needs, caregiver strain, and uneven access to geriatric and supportive services across the country.

**Public health implications—What are the key implications or messages for practitioners, policymakers and/or researchers in public health?**
Calls for evidence-informed interventions, including caregiver support programs, gender-sensitive policies, strengthened geriatric services, and community-based care models to address gaps in access, equity, and caregiver well-being.Identifies critical research and policy priorities, including intersectional and multilingual approaches, improved monitoring of caregiving outcomes, and expanded study of transnational elder care among Nigerian diaspora communities.

**Abstract:**

This integrative review on patterns of elder caregiving in Nigeria synthesizes evolving dynamics and determinants of caregiving practices amid demographic and household change. The objective of this review was to identify prevalent patterns of elder caregiving, explore the roles and responsibilities of caregivers, and examine the challenges and support needs within the Nigerian context. Academic Search Complete, CINAHL, PubMed, PsycINFO, and Medline were searched in November 2024. Inclusion criteria were peer-reviewed journal articles published in English focusing on elder caregiving among Nigerians; non-peer-reviewed sources (e.g., dissertations, conference papers, and books) were excluded. Data extraction was performed using a structured matrix, and findings were synthesized thematically. Risk of bias was appraised using SANRA (for narrative reviews) and MMAT (for empirical studies). Twenty studies published between 1991 and December 2022 were included. Analyses were guided by an intersectional conceptual framework spanning five domains: cultural, familial, economic, psychosocial, and policy. The interconnected dimensions illustrate how cultural expectations shape family caregiving roles, which in turn influence economic strain, emotional well-being, and access to institutional support. By emphasizing the interaction among gender, class, and social location within these domains, the framework demonstrates how caregiving operates as a multidimensional and relational process. Thematic synthesis identified six overarching themes: cultural influences, gender differences, family dynamics, economic factors, challenges faced by Nigerian caregivers, and government policies and support. Limitations include reliance on single-reviewer screening and extraction, exclusion of unpublished and non-peer-reviewed sources, restriction to English-language studies, and a focus on the Nigerian context, which may limit generalizability. Findings underscore that elder caregiving in Nigeria is multifaceted and shaped by intersecting gendered, cultural, and economic forces. Policy and practice should prioritize caregiver supports, accessible geriatric services, and gender-sensitive interventions, while future research applies the framework to address gaps in transnational and multilingual evidence.

## 1. Introduction

World Health Organization projects that by 2030, 1 in 6 people worldwide will be 60 years or over [1]. Globally, populations are aging rapidly, with the 60+ group expanding across regions [2]. Comparatively, Africa had a smaller proportion of elderly individuals in 2020 (5.6%) compared to other regions, such as North America (23.4%) [3]. Nonetheless, the elderly population in Africa continues to grow, and projections suggest that by 2050, Africa will have around 235 million elderly individuals. These shifts underscore urgent needs for family- and system-level responses in African contexts.

Nigeria is a Federal Republic consisting of 36 States and the Federal Capital Territory, Abuja. Administratively, states are grouped into six geopolitical zones (Figure 1) [4]. In Nigeria, the older population was estimated to be 11 million in 2020, making it the largest in any African country [5]. By 2050, it is projected that the number of older Nigerians will triple to over 33 million, representing roughly one in ten Nigerians [3,6]. This indicates that elderly individuals will account for approximately 10% of Nigeria’s total population by 2050 [6]. Nigeria has experienced increases in life expectancy at birth; by 2022, it reached 55.44 [7].

The increasing aging population in Nigeria will have implications for elder caregiving, as there will be a greater need for support from both expert and family caregivers [8]. Understanding aging in Africa is crucial for informing government policies and regional social and economic development [3]. However, despite the growing number of elderly individuals in developing countries like Nigeria, there is a lack of formal support arrangements and inadequate infrastructure to address the emerging issues in aged care [9]. Prior research highlights wide-ranging social, economic, and health consequences of population aging [10,11,12], but Nigeria-specific caregiving implications remain unevenly synthesized.

The concept of aging in Nigeria is multifaceted, encompassing both biological and social aspects, and holds important implications for society, community, and culture [9,13]. Traditionally, elder caregiving in Nigeria has been the responsibility of the family, with adult children and relatives fulfilling the caregiving role. The extended family system, deeply rooted in Nigerian culture, has historically provided support and care for elderly individuals. However, societal changes such as urbanization, migration, and modernization are impacting the traditional family structure and are leading to new patterns of elder caregiving [14,15].

The traditional Nigerian family structure was characterized by patriarchy and collectivism, where communal harmony and group solidarity took precedence over individual goals [13,16,17]. Kinship and community held great importance, and the traditional family unit provided care and support for elderly members, including food supply, welfare, and security [13]. Elders were highly respected, as they were considered the custodians of traditions, culture, and a source of wisdom. In the traditional Nigerian society, the community relied on elders for guidance, knowledge, and direction [18].

The impact of globalization on traditional African cultures has led to cultural imperialism, where foreign cultures dominate indigenous cultures in both material and non-material forms [19]. This cultural imperialism has had adverse effects on the indigenous family structure, traditional healthcare systems, and pre-colonial economies in Africa [19]. Consequently, in Nigeria, the traditional extended family system, which historically emphasized the importance of respecting and supporting older individuals, is gradually diminishing. This transformation can be attributed to factors such as the prevalence of nuclear family structures, economic development, labor migration, urbanization, industrialization, shifts in family dynamics, and the influence of new religious beliefs [9,13,19]. These shifts reshape who provides care, how care is financed, and where care occurs.

### Problem Formulation

Elder caregiving, also referred to as elder care, elderly caregiving, or care of older adults, involves providing physical, emotional, financial/economic support and social support to elderly individuals who require assistance due to age-related limitations or health issues. The aging population is a global phenomenon, and Nigeria, like many other nations, is undergoing a demographic transition towards a larger elderly population. This transition presents challenges for families and communities in terms of providing adequate care and support for older individuals [6]. Gaining insights into the changing patterns of elder caregiving in Nigeria is essential for the development of effective policies, interventions, and support systems to meet the needs of the aging population. Yet to date, the paucity of literature focused on the Nigerian context has resulted in limited synthesis on how intersecting factors (gender, socioeconomic status, migration, locality) shape caregiving patterns.

The integrative review examines the current landscape of elder caregiving in Nigeria, including the obstacles, gaps, and emerging trends. It highlights the factors that influence elder caregiving patterns, such as cultural norms, socioeconomic factors, and migration patterns. The review investigates the roles and responsibilities of various caregivers, including family members, formal caregivers, and community-based support systems. With the aging population growing and social dynamics evolving in Nigeria, comprehending the patterns of elder caregiving is essential for addressing the unique needs and challenges faced by elderly individuals and their caregivers. By conducting an integrative literature review, a comprehensive analysis of elder caregiving in Nigeria is presented. However, prior reviews have not explicitly applied an intersectional lens to synthesize how these factors interact, nor have they addressed transnational caregiving among Nigerian diasporas.

The overarching question guiding this review is: What are the current patterns of elder caregiving among Nigerians? This integrative review synthesizes Nigerian elder-caregiving research using a consistent intersectional framework to clarify who provides care, under what constraints, and with what consequences, and it foregrounds the underexplored transnational dimension as a priority for future work. To guide synthesis and enhance policy relevance, this review was guided by an intersectional conceptual framework organized around five interrelated caregiving domains: cultural, familial, economic, psychosocial, and policy.

## 2. Materials and Methods

An integrative review design was employed, as this design allows for the integration of diverse methodologies, providing a comprehensive synthesis of the literature [20]. A theoretical orientation guided the interpretation of evidence to promote coherence and depth across diverse sources [20]. Data were examined and organized thematically to identify patterns, relationships, and knowledge gaps. Following the methodology proposed by Whittemore and Knafl, the data were systematically reduced, displayed, compared, and analyzed to draw conclusions and ensure the accuracy of the findings [20].

Eligibility criteria included peer-reviewed studies in English focused on elder caregiving among Nigerians. Dissertations, books, conference papers, and other non-peer-reviewed sources were excluded. Included studies employed diverse designs such as qualitative, quantitative, mixed-method, and narrative approaches, examining caregiving roles, motivations, and contextual factors like gender, culture, and family structure.

The search was conducted across five databases (Academic Search Complete, CINAHL, PubMed, PsycINFO, and Medline) in November 2024. Search terms combined words related to aging, Nigeria, and caregiving and yielded approximately three thousand records before screening (Appendix A). A PRISMA flowchart [21] was used to organize the selection process and enhance transparency of the steps (see Appendix A). This review was not registered, and no protocol was prepared.

Titles and abstracts were screened, and full-text articles were assessed against eligibility criteria in consultation with a trained academic librarian. Duplicate records were removed using EndNote 20 [22], and all screening and full-text assessments were conducted manually. Screening and extraction were conducted by the first author, with reflective documentation maintained to ensure transparency. Data extraction followed a reporting matrix to ensure consistency in capturing study details, including author, year, design, population, methodology, and outcomes related to caregiving roles and experience (Appendix A). Quality appraisal was guided by the Scale for the Assessment of Narrative Review Articles (SANRA) framework for narrative studies (Appendix A) and the Mixed Methods Appraisal Tool (MMAT) for empirical studies (Appendix A). Findings were thematically synthesized to emphasize contextual interpretation, consistent with the methodological principles of an integrative review.

### 2.1. Theoretical Perspectives

The review is guided by an intersectional perspective, allowing for the critical exploration of how cultural, social, and gendered dimensions intersect to shape caregiving practices among Nigerians. This interpretive stance supports a holistic understanding of the complex realities surrounding elder caregiving within varying national and transnational contexts.

Intersectionality, as posited by Kimberlé Crenshaw, emerged from Black feminism and Critical Race Theory [23]. Since its inception, the framework has expanded and evolved across disciplines, aiming to understand how various social categories such as gender, race, class, age, and other intersecting differences shape and impact individuals and communities. Intersectionality recognizes the simultaneous experiences of multiple identities within specific spatial and temporal contexts throughout everyday lives [24]. This lived experience perspective highlights the complexity and fluidity of social categories and their interdependence.

By adopting an intersectional perspective, the review explores the interconnections between individual experiences and the intersecting systems of marginalization. Drawing on the works of scholars such as [23,25,26], the application of the intersectional lens supports insight into how multiple social categories, such as gender, age, socioeconomic status, and ethnicity, interact and shape the experiences of elder caregiving.

This lens acknowledges that individuals do not experience caregiving in isolation but are situated within broader social structures and power dynamics. It recognizes that different forms of oppression and privilege can intersect and compound, resulting in unique and complex experiences for caregivers and care recipients [27]. Through the lens of intersectionality, the study aims to uncover the nuanced connections and dynamics that exist within the realm of elder caregiving in Nigeria, ultimately contributing to a more comprehensive and inclusive understanding of the topic. For this review, intersectionality is used as both a theoretical and interpretive framework to understand how structural and cultural factors shape caregiving experiences. Through the intersectionality framework, the analysis examines the lived realities of social expectations and vulnerabilities experienced by elders and by informal caregivers such as spouses and adult children. This approach enables the comprehension of the intricate complexities and challenges encountered by caregivers as they navigate their caregiving responsibilities. Furthermore, the intersectionality perspective situates old age and caregiving within the broader socio-political and cultural structures that influence vulnerability across the life course. By broadening the understanding of intersectionality in the context of elder caregiving, it will be possible to gain valuable insight into the experiences of the caregivers and elderly individuals. These insights may inform targeted support programs and policies tailored to address the unique needs of elderly individuals [28,29].

### 2.2. Eligibility Criteria

Inclusion criteria specified the studies: (1) Articles published prior to December 2022 in peer-reviewed academic journals written in English. (2) Studies focusing on elder caregiving among Nigerians. (3) Studies focusing on caregiving dynamics, experiences, and perspectives of either caregivers or care receivers. (4) Studies following quantitative, qualitative, or mixed-methods study design or using any type of review design. To enhance credibility, commentaries and opinion papers without empirical data were excluded due to their reliance on personal viewpoints. Dissertations, conference papers, conference abstracts, books, and book chapters were also excluded.

To expand the search and make it comprehensive, an in-depth search was performed in consultation with a content-expert librarian. The Boolean literature search terms included a combination of subheadings (MeSH) and keywords: Aged* or older adult* or older people* or elderly* or senior citizen* or ag* or gerontology or geriatrics AND Nigeria* or Nigeria, Eastern* or Nigeria, Western* or Nigeria, Southern or Nigeria, Northern** or Nigerian, Overseas*, Nigerian, abroad* AND Care or caregiv* or support.

Searches were executed by the first author in collaboration with the content-expert librarian; no automation tools were used to develop or run the searches. All sources were last searched in November 2024. No database limits were applied at the search stage beyond standard indexing; English language and peer-reviewed status were enforced during screening according to the inclusion criteria. Reference lists of included studies were hand-searched to identify additional relevant sources.

### 2.3. Literature Search Stage

Multiple databases were systematically searched from inception for peer-reviewed articles focused on elder caregiving. Various sources of information, including professional journals, periodicals, books, internet sources, dissertations, and gray literature, were searched; however, non-peer-reviewed sources, books, dissertations, and gray literature were excluded at the eligibility stage. These sources were accessed through the following databases: Academic Search Complete (Ebsco), CINAHL (Ebsco), Medline (Ovid), PsycINFO (Ebsco), and PubMed (Medline). A backward citation hand search was also conducted. The initial literature search of 5 databases yielded 2986 citations. Duplicate articles (*n* = 1668) were detected and removed using a bibliographic management system (EndNote). The initial screening involved assessing titles and abstracts of 1318 articles, resulting in the elimination of 1280 articles that did not meet inclusion criteria. Screening of titles, abstracts, and full texts was conducted by a single reviewer, and no automation tools were used in the selection process (see Appendix A).

The remaining 38 articles underwent full-text review. Twenty-three were excluded, including 11 reflection papers and 12 that did not align with the main subject in terms of population, study focus, and outcomes. This left 15 articles from the database search. Additionally, 14 articles were identified through a backward citation search and underwent full screening. Nine were excluded (one reflection paper and eight that did not meet inclusion criteria), resulting in 5 articles retained. In total, 20 articles were included in the final synthesis (15 from the database search and 5 from the backward citation search). Potential missing or unpublished studies were acknowledged as limitations due to the small number of eligible papers.

### 2.4. Data Extraction and Quality Appraisal

A data extraction matrix was created to provide an organized overview of the identified twenty studies (Appendix A). The matrix served as a tool for sorting and categorizing the different arguments and findings related to the research question [30]. Data extraction included details such as author, journal, year of publication, population, objectives, study design, methodology, data collection methods, and findings (see Appendix A). This structured approach facilitated the synthesis of information in a transparent and reproducible manner.

Two types of quality assessment tools were employed depending on the methodology of the included article: the SANRA [31] (Appendix A) and the MMAT [32,33], applied separately to qualitative (Appendix A), mixed-methods (Appendix A), and quantitative studies (Appendix A). The SANRA scale, consisting of six items rated from 0 (low standard) to 2 (high standard), evaluates articles based on the importance and aims of the review, the literature search, referencing, presentation quality, level of evidence, and relevance of endpoint data [31]. The MMAT Version 2018 [34] recognizes distinct criteria for assessing quality in qualitative and quantitative methods, considering their different assumptions and objectives. The MMAT utilizes appraisal ratings that are classified as Yes, No, or Cannot Tell. In this review, the interpretation of quality assessments followed [20]’s recommendation of applying a simplified two-point scale: studies with lower appraisal ratings (equivalent to 2 or below) were considered lower quality, whereas those with higher ratings (3 or above) were considered higher quality. The quality appraisal tools were used to inform the analysis by identifying any possible discrepancies in findings. Evaluating data quality focused on assessing how each study contributed to understanding caregiving contexts rather than assigning scores.

The question guiding this review focused on the patterns of elder caregiving among Nigerians, both those living in Nigeria and abroad. Thematic analysis was used to identify and interpret recurring ideas, relationships, and contextual influences in the selected studies. This approach allowed for a flexible and reflective interpretation of the literature, emphasizing meaning and context rather than measurement. Themes were grouped following the principles of reflexive thematic analysis [35] and integrative synthesis [36]. Data were synthesized thematically in keeping with the integrative review approach to capture patterns across diverse study designs and contexts. Given the heterogeneity of study designs, data types, and analytical approaches, statistical synthesis was not feasible or appropriate.

### 2.5. Data Analysis

In total, twenty codes were identified during the analysis of the findings from the selected studies. These codes were then categorized into six themes based on their relevance to the research question. The themes provided a framework for organizing and interpreting the data. The findings were examined and coded manually by the reviewer according to the description provided for each theme. This process helped to identify key patterns, similarities, and differences in the data across the studies. Data analysis focused on exploring relationships between themes and how these collectively shaped understanding of elder caregiving within Nigerian contexts. This stage emphasized reflective interpretation, allowing patterns to emerge organically from the reviewed studies.

## 3. Results

### 3.1. Study Characteristics

This review synthesizes findings from 20 studies published between 1991 and 2025 with a range of research methods, including qualitative methods (*n* = 8), mixed methods (*n* = 5), quantitative methods (*n* = 6), and a narrative review (*n* = 1).

Participants included elderly individuals (*n* = 1354), care partners (*n* = 3850), educators (*n* = 684), and government officials (*n* = 3), enabling analysis across caregiving roles, generational positions, and gender identities. This diversity allowed for interpretation of how caregiving responsibilities are distributed and experienced differently across social groups. Although the studies covered multiple regions, representation was uneven, with most concentrated in the South West, South East, and South South zones and limited evidence from the North Central, North East, and North West.

### 3.2. Themes and Sub-Themes

The thematic analysis highlighted similarities and distinct differences among the studies shaped by intersecting social identities and structural conditions. Six interrelated themes emerged, capturing the complexity of elder caregiving practices in Nigeria as shaped by overlapping cultural, gendered, economic, and institutional factors: cultural influences, gender differences, family dynamics, economic factors, challenges faced by Nigerian caregivers, and policy or institutional support. Together, these themes demonstrate how caregiving is not a uniform experience but one mediated by intersecting forces such as gender, class, marital status, regional context, and access to formal support. The synthesis highlights both the endurance of cultural caregiving norms and the adaptive strategies caregivers employ in response to shifting economic realities, migration, and institutional neglect. These adaptations are unevenly distributed, with caregiving burdens falling disproportionately on women, low-income families, and rural communities. As shown in Table 1, the reviewed studies collectively reveal that while caregiving practices remain deeply rooted in cultural and familial obligations, gender, economic strain, and shifting family structures intersect to shape caregiving outcomes. These interrelated factors highlight both the resilience of traditional caregiving systems and the emerging vulnerabilities within them.

### 3.3. Cultural Influences

This theme, addressed in 10/20 articles, explores how cultural norms, traditions, and moral values shape caregiving expectations and practices across Nigerian communities [8,18,37,38,39]. Across these studies, caregiving norms intersect with gender, marital status, and economic position, shaping who provides care, how it is valued, and the degree of support available to the elderly. For example, moral and religious values reinforce caregiving roles [40], yet these expectations can become burdensome for female caregivers in rural areas with limited resources [41].

Culture shapes attitudes, obligations, and role expectations in caregiving, and these norms are reproduced through moral, religious, and economic logics that vary by region and class [40]. For instance, old age in the Esan community is framed through ideals of purity and virtue, which shape expectations around elder respect and care. Traditional Esan society operates on intergenerational exchanges of wealth and care, where children provide support to their elderly parents. This model reflects a cultural logic of reciprocity and lineage continuity, but its enactment depends on the caregiver’s gender and economic capacity.

Caregiving motivations reflect overlapping social roles and structural constraints. Spouses emphasized emotional commitment and marital duty, while adult children highlighted filial piety and the moral expectation to reciprocate parental care; these motivations are mediated by class and gender, as those with greater financial means could express care through remittances or housing support, while others relied more on emotional and physical caregiving [14,37,38,40].

Given Nigeria’s ethnic and regional diversity, caregiving operates within layered systems of identity, tradition, and inequality. Across various cultures in Nigeria, there is a shared value of respecting and honoring elders, although the form and intensity of caregiving differ across regions, shaped by intersecting factors such as ethnicity, religion, and class [18,42]. It is considered a moral duty for younger family members to provide care and support for their aging parents or relatives [39]. This cultural value often drives caregiving practices within families [43]. Filial piety, central to many Nigerian ethnic traditions, intersects with patriarchal norms and economic constraints, often placing disproportionate caregiving expectations on women. Filial piety is highly valued in Nigerian society, and children are expected to fulfill their responsibilities by providing care for their aging parents [44,45]. Failing to meet these expectations may result in social disapproval, particularly for daughters or firstborns who deviate from traditional caregiving norms [14,15,39,43,46]. Care obligations include financial assistance, emotional support, and physical caregiving, but the ability to fulfill these roles is shaped by intersecting factors such as gender, class, and migration status [8,37,38]. The cultural value placed on filial piety influences caregiving practices within Nigerian families and highlights the importance of intergenerational support and care. However, the burden of fulfilling these expectations is unevenly distributed by gender, birth order, and socioeconomic status, often falling on women with limited resources.

In study [46], the authors found that the living arrangements of elderly individuals varied across different geopolitical zones in Nigeria. Forms of support are stratified by class and region, indicating that spatial variation interacts with class and services: while wealthier families often provide formal or financial assistance, lower-income families depend on collective caregiving through shared housing and communal reciprocity [8,37,38]. These findings reveal that caregiving is not simply a moral obligation but a socially embedded practice shaped by intersecting systems of gender, class, kinship, and regional inequality.

### 3.4. Gender Differences

This theme, addressed in 10/20 studies, examines how caregiving roles and expectations are structured by gender and how these intersect with class and location [6,39,43,46,47]. Gendered caregiving norms in Nigeria produce persistent inequalities, as caregiving expectations intersect with gender, class, and geography to create uneven caregiving burdens and access to support. Caregiving was framed as a feminized moral duty with women’s identities closely tied to care roles; these expectations intersect with poverty and limited formal support, intensifying women’s burden [41,43,46,47,48]. For instance, ref [39] reported that 60% of caregivers were female, while [47] identified women as the primary caregivers in most households, confirming a dominant pattern of female-led care [43,48]. While these findings affirm caregiving as a gendered expectation, they also obscure structural constraints such as poverty and lack of formal support that intensify caregiving burdens for women.

Adult daughters generally exhibit more positive attitudes toward caregiving and are less likely to perceive it as burdensome compared to adult sons [48]. This contrast reflects how gendered socialization positions women as emotional caregivers and men as financial contributors but also reveals how caregiving labor is unequally valued. Studies diverge on the extent to which these divisions remain stable. Research has described persistent female centrality [39,48], while other studies note that education, urbanization, and migration are producing incremental role fluidity, with men increasingly participating in emotional or financial care [6,47]. This demonstrates that gender norms operate along a continuum rather than a fixed binary.

In Nigerian culture, adult daughters may face challenges in openly expressing their emotions but prioritize their sense of duty. Women’s pursuit of education intersects with caregiving expectations, often creating conflict between professional aspirations and familial obligations [6]. As modernization promotes individual advancement through education, this shift can result in reduced availability of care and support for elderly individuals, particularly in contexts where caregiving is culturally feminized. Across studies, education emerges as both a liberating and destabilizing factor: it enhances women’s earning power and decision-making [48] but also creates time conflicts that reduce hands-on care [6,39]. This tension illustrates intersectionality in practice, as class mobility reconfigures but does not eliminate gendered obligation. Women are more likely to experience financial limitations and face psychosocial health challenges in older age compared to men [43]. Additionally, females tend to experience chronic ailments at an earlier age than males [39], suggesting that gendered care burdens accumulate into health disadvantage over time. These findings reveal layered inequalities. Women’s dual role as caregivers, often responsible for both older and younger family members, and as aging dependents themselves creates a cyclical inequality. Those who provide the most care receive the least support in later life, while men’s greater economic security buffers their dependence in old age [39,43].

A gender imbalance in providing assistance to the elderly has been observed, particularly in food remittances, where elderly women receive more support [42]. This indicates that gender also shapes the type and value of care elders receive. Where economic dependence is high, women’s care work and vulnerability often coexist, raising risks of neglect or exploitation [43]. In sum, gendered patterns of care cannot be interpreted in isolation from class and location. Urban educated women negotiate care through paid services or remittances, while rural low-income women provide direct care with limited support [6,48]. Such intersectional differences explain why statistical gender effects may appear muted even as qualitative accounts reveal persistent asymmetry.

Age and gender influence the level of support individuals receive, particularly in rural areas [49]. Intersectional factors such as rural residence, widowhood, and class position shape caregiving outcomes in distinct ways. Older women in urban areas, especially widows, are at higher risk of isolation, while some rural men mitigate vulnerability through remarriage [49]. The study also found that rural women under 75 were more likely to receive gifts from their children, whereas elderly fathers received more substantial financial support [49]. These patterns show how gender interacts with age and geography to produce layered reciprocity, where women receive frequent low-value support that maintains emotional ties, while men receive fewer but higher-value transfers linked to status and decision power.

Awareness of elderly individuals’ needs also varies by gender. Across four Nigerian geopolitical zones—South South, South East, North Central, and South West—men generally demonstrated greater awareness of the needs of elderly individuals in their communities [39]. This trend was particularly evident in the South South, South East, and North Central regions. However, in the South West zone, women showed higher levels of awareness than men. According to [39], these gender differences may stem from men’s greater engagement with their surrounding environments, which enhances their perception of community needs. Regional differences alongside gendered care roles can be greatly affected by local social organization. Such differences expose the uneven geopolitical representation of existing studies, as few studies originate from the North East and North West, limiting comparability and generalization of findings at a national scale in Nigeria. Overall, gender continues to organize caregiving labor, yet its expression is mediated by education, income, and urban residence.

### 3.5. Family Dynamics

This theme, addressed in 12/20 studies, focuses on how family structures and intergenerational relationships shape the organization of elder caregiving. Across studies, the family remains the primary source of elder care in Nigeria, reflecting shared moral and relational obligations that sustain intergenerational continuity [6,38,39,40,46,48,49]. Immediate family members, including adult children, spouses, and siblings, most often assume caregiving duties, reflecting shared moral and relational obligations within households. Reciprocity and intergenerational exchange remain central, yet the strength of these ties differs by region, class, and migration status [37,39]. However, caregiving roles are shaped by intersecting social identities and structural conditions. Studies converge on the idea that family care is morally obligatory but diverge on whether this norm is still sustainable under contemporary socioeconomic pressures [6,14].

Shifts in social organization have redefined traditional caregiving arrangements [15]. This outsourcing trend signals class differentiation, as middle-class urban families substitute kin care with paid support, whereas rural families preserve mutual aid through communal labor [14,42]. Marital conflict, sibling tension, and limited support from male children intensify the social and emotional strain experienced by both elders and their caregivers [44]. Caregiving is thus negotiated within families along gendered and generational lines, and stress is distributed unevenly depending on social position.

Caregiving responsibilities within the family are typically shared among family members, although some individuals may bear a greater burden than others [40,47]. The family’s ability to meet these obligations directly affects elders’ well-being and quality of life [38,41]. Studies diverge on how “shared” this care actually is. Ref [47] reported gender-balanced contributions in dual-income households, while [40] found persistent female overrepresentation in unpaid care. This variation reflects intersectional differences, as education, class mobility, and urban residence reconfigure traditional caregiving hierarchies.

Intergenerational reciprocity is a defining feature of Nigerian family caregiving. Elderly individuals who were once caregivers themselves are now relying on their children and grandchildren for care as they age [37,44]. Married children often have elderly female relatives living with them to receive assistance, guidance, and support [15]. This reciprocal pattern aligns with descriptions of filial care as both a moral duty and an investment in future reciprocity [38]. Yet, other studies reveal tension when economic constraints limit children’s capacity to reciprocate, creating guilt and emotional strain [6,39]. These findings reflect continuity and adaptation in caregiving norms shaped by migration, financial limitation, and evolving family configurations.

Societal transformations, including international migration, urbanization, industrialization, Western influence, and the transition to a nuclear family structure, have been recognized as influential elements that have affected the extended family system in Nigeria [6,14,40,50]. Earlier research portrayed these changes as cultural decline [15,51], whereas more recent studies interpret them as structural adaptation, where caregiving is redistributed through remittances and emotional support [50]. These adaptations intersect with gender, class, and marital status, as urbanization may expand access to resources while simultaneously weakening kin-based safety nets.

Migration alters emotional and relational aspects of caregiving [42,50]. Elderly parents with migrant children often experience loneliness and reduced daily interaction despite financial remittances. The absence of younger relatives limits hands-on assistance, and these effects vary by gender and class, as widowed or low-income women are more vulnerable, while male elders with property or pensions experience greater stability [14,49]. These relational disruptions affect Activities of Daily Living (ADLs) and Instrumental Activities of Daily Living (IADLs), illustrating how family separation reshapes both practical and emotional caregiving.

Research indicates that filial piety has weakened in some contexts [14,15,40]. Younger generations are less willing to sacrifice employment or migration opportunities to provide hands-on care, reflecting tensions between modern aspirations and cultural duty. Conversely, rural and conservative communities maintain enduring reciprocity, anchored in lineage systems and land inheritance that favor male elders [49]. These contrasts demonstrate that modernization reshapes rather than erodes filial duty, with outcomes contingent on income, kinship capital, and geography.

Place of residence interacts with gender, class, and kinship to shape elder-care outcomes. Elders in urban areas are more likely to receive consistent support [38], yet that support may be more financial than personal, reflecting middle-class time scarcity. Rural elders receive hands-on care but fewer material resources, highlighting the trade-off between presence and provision [6,15,49]. These patterns reveal deep structural inequalities, as access to care is stratified by income, housing stability, and proximity to kin [6,14,40]. Overall, family caregiving in Nigeria functions as an adaptive institution where enduring cultural norms intersect with gender, class, and geography, producing varied and sometimes conflicting outcomes. The persistence of family-based care underscores its cultural significance, yet its form and intensity are increasingly stratified by economic capacity, gender, and migration, illustrating how structural inequalities shape caregiving relationships.

### 3.6. Economic Factors

This theme, addressed in 11/20 studies, examines how financial capacity, employment, and class inequality determine access to, and quality of, elder care. Financial limitations and scarce resources directly shape the availability, type, and quality of elder care, often in ways that reflect broader inequalities [43,46,48,52]. Families may face challenges in affording necessary medical treatments, assistive devices, and home care services, which can lead to increased caregiver burden and inadequate support for the elderly [8,44,53]. Studies consistently show that the quality and consistency of elder care are stratified by financial capacity. Wealthier families have the means to provide comprehensive care, including personal doctors and caregivers, while poorer families and communities may only be able to provide limited care in terms of basic needs such as feeding, medical attention, clothing, and affection [45]. Across studies, the divide between resource-rich and resource-poor households is a consistent finding, yet the mechanisms differ: Household income and employment have been emphasized as key factors influencing caregiving [46,48], while other research attributes disparities to state underfunding of geriatric care [43]. These disparities are not solely economic but also intersect with gender and geography. For example, low-income women caregivers in rural areas may face compounded burdens due to limited infrastructure, unpaid labor expectations, and lack of access to formal support [45,50].

Elderly individuals employed various coping strategies to supplement the insufficient support provided by their children [14]. Older men tended to rely on pensions, farming, and offspring contributions, whereas women engaged in small trading, informal labor, or charity to survive [14]. These contrasting strategies reveal how gendered economic inequality extends into old age, as men benefit from asset ownership or pensions tied to formal employment, while women depend on informal and unstable income streams. Such divergence underscores the cumulative effect of gendered labor histories and class position across the life course

Migration, whether internal or international, driven by economic hardship, unemployment, and Westernization, reshapes family caregiving dynamics in complex ways [14,15,42,46,50]. For some families, remittances from migrant children help offset caregiving expenses and improve elders’ material well-being. However, for others, the departure of younger relatives creates new costs for substitute care and reduces shared household resources, intensifying financial strain. Migration therefore brings both relief and pressure, as economic gains often coexist with the erosion of local support systems [42,50].

Care obligations towards elderly parents are often renegotiated based on the economic opportunities and constraints faced by migrating children, leading to selective caregiving practices. In certain cases, there is a noticeable shift away from traditional Nigerian caregiving values rooted in love and empathy. This shift reflects an increased focus on material gain, sometimes diminishing empathy and increasing the risk of neglect or mistreatment of elderly family members [43]. Overall, the studies show that economic pressures and cultural expectations intersect to reshape caregiving norms, privileging elders with financial and social capital while marginalizing those without such resources.

### 3.7. Caregiver Strain and Psychosocial Dimensions

This theme, addressed in 7/20 studies, captures the emotional and psychological consequences of caregiving, focusing on stress, burnout, and coping responses. Caregiving challenges are described not only as structural but also as deeply emotional and relational [8,43,44,53]. The most consistent finding was emotional fatigue and burnout resulting from sustained caregiving with little respite or support [43,53]. Prolonged caregiving without institutional aid leads to physical exhaustion, irritability, and depressive symptoms [8,43,44,53]. The absence of formal respite services was identified as a contributing factor to caregiver fatigue [43], while socioeconomic and gendered pressures were also identified as key drivers of emotional strain that accumulate at the intersection of gender, class, and caregiving duration [8,53]. Together, these studies demonstrate that burnout is a widespread outcome of long-term caregiving, transcending geographic and socioeconomic contexts.

Caregivers consistently reported stress from balancing employment, domestic responsibilities, and elder care [6,8,37]. These overlapping duties create continuous tension between economic survival and family expectations. Ref [47] described differing experiences among caregivers: some adapt through informal delegation and cooperation within households, whereas [44] noted caregivers experience emotional strain from unequal role sharing, with women taking on more care work despite men’s financial authority. When examined through an intersectional lens, this contrast underscores how coping and burden coexist along a continuum shaped by gender, class, and access to social support. Collectively, they illustrate that caregiving operates within overlapping systems of expectation and constraint that distribute emotional and physical burden unequally.

Caregivers frequently experience emotional fatigue and isolation when coping resources are limited [43,44,46,48]. Cultural norms of endurance and self-sacrifice often discourage open expression of distress, leading caregivers to suppress emotional pain and internalize stress. Education and access to social networks can reduce loneliness [48], while regional variation in community engagement suggests that isolation may be mitigated where communal ties and caregiver networks remain strong [46]. When examined from an intersectional perspective, caregiver strain emerges not only from the intensity of care tasks but also from the interaction of emotional invisibility, gendered expectations, and unequal access to coping support.

### 3.8. Government Policies and Support

This theme, addressed in 9/20 studies, examines the role of government policy and institutional frameworks in shaping elder care and access to formal support. Government policy, geriatric health services, and formal supports remain limited and fragmented in their capacity to meet the complex needs of Nigeria’s aging population [38,41,43,45]. They emphasize the need for targeted social policies that account for the intersecting vulnerabilities of older adults, particularly those who are poor, female, or living in rural areas. This convergence reflects broad consensus that elder care is under-prioritized in national policy frameworks. The absence or weakness of support care services has resulted in neglect and delayed treatment for elderly people [41]. The absence of a coordinated social security or pension framework continues to place caregiving responsibility on families, reinforcing reliance on unpaid kin labor and limiting institutional accountability [38].

Poverty, social isolation, and limited access to care reflect the social consequences of weak geriatric services and institutional neglect [43]. Limited awareness and poor accessibility of formal support services further intensify the burden on families in underserved regions [41]. Rapid population aging has not been matched by the development of social or community-based care frameworks, leaving many older adults dependent on informal systems that vary by region [45].

At the system level, geriatric medical services remain insufficiently prioritized within the Nigerian health system, where long waiting times, a high provider–patient ratio, and poor communication between healthcare providers and older adults discourage service use and foster mistrust [43]. Limited policy attention to geriatric care functions as a structural mechanism of exclusion, as minimal state engagement amplifies class, gender, and geographic disparities rather than reducing them [43]. There is need to develop community-based care models to address service gaps [45] and to strengthen national policy reform to ensure more consistent and equitable elder care across regions [38]. Overall, the limited policy response reinforces dependence on unpaid family care, demonstrating how institutional neglect reproduces inequality in old age.

## 4. Discussion

Elder caregiving in Nigeria is shaped by intersecting cultural, gendered, and economic pressures that reflect broader structural inequalities. These intersections reveal how caregiving responsibilities are distributed and constrained by societal norms, migration patterns, and limited formal support. This analysis interprets caregiving through an intersectional lens, emphasizing how overlapping systems of gender, class, and social location shape who gives and receives care. Cultural values such as filial piety continue to define caregiving as both a moral and spiritual responsibility. However, intersectional analysis reveals that these expectations are unevenly distributed across gender and class. While filial piety reinforces intergenerational support, it simultaneously obscures the disproportionate burden placed on women with limited socioeconomic mobility [54]. The gendered overlap between paid employment and family caregiving illustrates how care work remains undervalued and structurally unsupported, reflecting gendered economic inequality [55,56]. Patriarchal norms rooted in religion and community values position women as natural caregivers, while men are primarily expected to provide financial or logistical support. These patterns align with research emphasizing the moral duty of children to honor their parents [57,58,59] but also reveal how moral duty translates into gendered labor expectations. This caregiving imbalance reflects broader African caregiving systems, where women disproportionately shoulder care responsibilities under the combined pressures of patriarchy and economic constraint [60,61].

Migration and urbanization further compound caregiving strain. As younger family members relocate for work, older adults are left behind with diminished support networks and must rely on informal coping strategies [62,63]. Economic constraints intensify these inequalities, as lower-income families depend on unpaid female labor and communal reciprocity, whereas wealthier households can outsource care responsibilities [46,48]. These structural shifts have reshaped rather than replaced filial obligations, as caregiving increasingly occurs through financial remittances rather than co-residence [9,13,19]. Together, these trends mirror broader African caregiving transitions, where migration and economic restructuring redefine intergenerational support systems within a shared model of family solidarity amid weak institutional support [60,64]. These patterns are also gendered, as women’s caregiving labor often sustains transnational families even when migration redistributes economic resources unequally. Caregiving outcomes emerge at the intersection of gender, class, and mobility. Migration often deepens class divides: some families gain resources to sustain elders from abroad, while others experience emotional distance and reduced hands-on support [50]. While women often receive greater informal or material support, these advantages do not necessarily translate into equitable access to formal services. This pattern suggests that gender interacts with economic status and place of residence to shape both the visibility and intensity of care, revealing how structural inequalities mediate caregiving experiences across contexts. The few available studies on transnational care show how migrants sustain moral responsibility despite spatial separation [63], indicating the need for broader inquiry into cross-border caregiving.

Government inaction compounds these inequalities in health access, financial security, and caregiving burden. Social protection for older adults remains minimal, and geriatric services are scarce or poorly coordinated [38,45]. Without institutional support, care continues to depend on family solidarity, leaving vulnerable elders without adequate health or income security. When situated within broader African scholarship, Nigeria’s experience aligns with regional patterns where family networks compensate for weak welfare systems, reflecting caregiving practices grounded in kinship, moral reciprocity, and communal responsibility [60,65].

Transnational caregiving remains underexplored despite its growing relevance in diasporic Nigerian communities. Although the National Policy on Aging was introduced, its implementation remains fragmented [66]. Geriatric services are under-resourced, and formal supports are inaccessible to many older adults [67]. Additionally, the exclusive use of English-language sources may have introduced linguistic bias, potentially excluding culturally nuanced caregiving perspectives rooted in indigenous Nigerian languages and oral traditions. This concern is echoed in scholarship that critiques Western-centric caregiving frameworks for marginalizing local epistemologies and lived realities [68]. Future research should examine with greater depth multilingual and community-based literature to describe with greater depth diversity within Nigeria’s caregiving realities.

The strength of this integrative review lies in its synthesis of diverse evidence and the application of an intersectional lens to contextualize caregiving patterns. However, several limitations must be acknowledged. The review is based solely on published English-language studies, which may have limited the representation of culturally embedded caregiving practices, particularly those rooted in oral and Indigenous knowledge systems. Risk of selective reporting could not be fully assessed, and the single-reviewer process introduces potential subjectivity despite careful documentation and review by co-authors. Reflective documentation and consultation with a content librarian helped maintain transparency and reduce bias during analysis. These methodological constraints may have influenced theme interpretation, particularly in areas with uneven regional data, thereby narrowing the diversity of perspectives represented. Future research should more explicitly apply intersectionality in empirical work, incorporate multilingual and community-based sources, and compare Nigerian caregiving patterns with those from other African and diaspora contexts. Such approaches would expand theoretical understanding and strengthen evidence for policy reforms that balance cultural continuity with equitable, gender-sensitive elder care.

This review advances an intersectional conceptual framework of elder caregiving in Nigeria, structured around five interrelated domains—cultural, familial, economic, psychosocial, and policy. These interconnected dimensions illustrate how cultural expectations shape family caregiving roles, which in turn influence economic strain, emotional well-being, and access to institutional support. By emphasizing the interaction among gender, class, and social location within these domains, the framework demonstrates how caregiving operates as a multidimensional and relational process. This intersectional model guided both the synthesis of findings and the development of targeted policy recommendations, offering a conceptual scaffold for future research, comparative studies, and policy design.

### 4.1. Implications for Policymakers, Practitioners, and Researchers

The implications for policymakers, practitioners, and researchers in elder caregiving in Nigeria are noteworthy and multifaceted. The findings emphasize the significance of targeted interventions, policy changes, and additional research to address the challenges experienced by caregivers and elderly individuals, thereby enhancing their well-being and the quality of care they receive. Advocacy efforts are essential in creating awareness among policymakers about the specific issues faced by the elderly and their caregivers, leading to the establishment and implementation of comprehensive welfare programs and policies that meet their needs. Effective policy responses should address the realities caregivers face, including financial strain, gendered caregiving roles, and limited access to formal support. Prioritizing income assistance, affordable healthcare, and flexible employment options can help mitigate these pressures and improve care outcomes.

Researchers play a central role in expanding the evidence base on elder caregiving. Future research should move beyond descriptive accounts to adopt intersectional and comparative frameworks that capture the combined effects of gender, class, and cultural context on caregiving practices. Mixed-methods and community-based designs are particularly valuable for documenting diverse caregiving realities across Nigeria’s regions. Researchers should include multilingual data sources to reduce linguistic bias and ensure inclusion of caregivers whose voices are often excluded from English-language studies.

Practitioners in the field of elder caregiving in Nigeria play a crucial role in supporting and assisting caregivers. They have a range of responsibilities that contribute to the well-being of both caregivers and the elderly. Practitioners offer caregivers valuable support through education, emotional assistance, access to resources, care planning, coordination, and advocacy. They provide training to enhance caregiving skills, guide caregivers in managing their emotions, connect them with community services, develop personalized care plans, and advocate for their needs. Embedding intersectionality into practice is essential to recognize how gender, class, and cultural context shape caregiving experiences, particularly among low-income and rural caregivers. Collaborating with faith-based organizations, NGOs, and other stakeholders allows practitioners to leverage resources and expertise, expanding the scope and effectiveness of services for the elderly. Their crucial role empowers caregivers and promotes the overall well-being of both caregivers and the elderly.

In summary, practitioners, policymakers, and researchers play a vital role in enhancing the well-being and support available to the elderly population in Nigeria through policy advocacy, capacity building, collaboration, service delivery, and research. Integrating intersectional and gender-sensitive frameworks into these processes ensures that interventions are responsive to diverse caregiving realities. By actively addressing these areas, all stakeholders can contribute to creating a more inclusive and supportive environment for elderly individuals and their caregivers.

### 4.2. Key Recommendations

Based on the findings of this review, the following key recommendations are made to improve elder caregiving in Nigeria:Develop comprehensive support systems: Government agencies should prioritize the development of comprehensive support systems that address the challenges faced by caregivers, including conflicting responsibilities, limited healthcare access, and a lack of formal support. A National Caregiver Support Package can include care navigation embedded in primary health centers to link caregivers to available benefits and services. Respite programs can be made accessible through registered community providers, while transport and medication subsidies can be allocated based on standardized ADL and IADL assessments. A 24 h caregiver helpline and a digital caregiver registry enable rapid referrals and visibility across service networks.Strengthen community-based initiatives: Community organizations and local interest holders may expand caregiver support through rotating respite services, peer groups, and targeted training. In rural areas, where caregiving often relies on informal reciprocity networks, state social welfare departments formalize these arrangements into Village Care Committees that can coordinate home visits, emergency relief, and clinic referrals. To reinforce these structures, caregivers are registered as community health auxiliaries, gaining access to basic training, referral tools, and stipends. Local governments allocate microgrants to caregiver-led cooperatives that manage transport, nutrition, and medication logistics for dependent elders. Public recognition through caregiver ID cards and inclusion in local planning councils positions caregiving as a civic contribution with structural support.Promote flexibility in cultural and gendered caregiving expectations: Challenging traditional gender roles in caregiving requires coordinated action across education, media, and labor systems. Awareness campaigns and school-based programs can be helpful to normalize shared caregiving responsibilities across genders through storytelling, peer dialogue, and community theater that reshape expectations. The Ministry of Labor can extend family-friendly workplace policies, including flexible scheduling, paid elder-care leave, and caregiving tax credits. These provisions explicitly recognize elder care and incentivize male participation, repositioning caregiving as a collective social responsibility rather than a default burden assigned to women.Foster intergenerational relationships: Strengthening ties between generations requires more than promoting caregiving demands; it requires creating conditions where younger adults see long-term value in staying rooted. Local governments can offer land access, housing credits, or business incubation support to youth who co-reside with older relatives, linking caregiving to economic opportunity. Intergenerational apprenticeship programs allow elders to pass on vocational skills, cultural knowledge, and community leadership, positioning older adults as active contributors rather than passive recipients. Community cooperatives that include both youth and elders in decision-making, resource sharing, and local enterprise development reinforce mutual investment in place. These strategies reduce the economic and social pull of urban migration and rebuild intergenerational trust through shared purpose.Advocate for policy changes: Coordinated reform requires sustained collaboration across research, practice, and governance. Elder caregiving must be formally embedded within national health and social protection frameworks, supported by dedicated budget lines and measurable implementation targets. Nigeria currently lacks a central coordinating body for elder care policy. To fill this gap, a National Commission on Aging and Caregiving can be established to align efforts across health, labor, education, and finance sectors, elevating elder care from a peripheral welfare concern to a national development priority. A National Elder-Care Fund could allocate resources to enhance access to geriatric units in tertiary hospitals and finance mobile outreach services for home-bound elders. Caregivers’ leave policies are embedded in public sector employment codes and incentivized across private industries through tax relief and compliance credits. Inclusion of elder care in the National Health Insurance Scheme enables caregivers to enroll dependents and access subsidized services. Annual reporting on caregiver support benchmarks and equity outcomes ensures accountability and drives continuous improvement.

Implementing these recommendations can enhance the well-being and support for the elderly population in Nigeria, creating a more inclusive and supportive environment.

## 5. Conclusions

The findings of this study underscore the multidimensional and intersecting nature of elder caregiving in Nigeria, highlighting the urgent need for comprehensive support systems and policies. Caregivers in Nigeria face a myriad of challenges, including conflicting responsibilities, limited access to healthcare, and inadequate formal support structures. To address these issues and ensure better care for the elderly, a holistic approach that integrates government, community, and cultural interventions is required. Government interventions are crucial in establishing policies that prioritize elder caregiving, provide financial support, and improve healthcare accessibility. Actions to establish policies that support elder caregiving include the need to expand financial support and strengthening healthcare accessibility. Community initiatives, such as the establishment of caregiver support groups and community-based care services, can enhance social support networks and alleviate the burden on individual caregivers. Additionally, cultural shifts are essential in challenging traditional gender roles and expectations, enabling more equitable distribution of caregiving responsibilities and fostering a supportive environment for caregivers.

The intersectional framework clarified caregiving dynamics and informed targeted recommendations, offering a foundation for future research and policy innovation. Recognizing the intersectional dynamics at play is pivotal in understanding the unique challenges faced by different groups of caregivers. It is crucial to adopt an inclusive approach that considers factors such as gender, socioeconomic status, and cultural norms to ensure that support systems are tailored to the diverse needs of caregivers and elderly individuals in Nigeria. Future research should expand beyond the boundaries of Nigeria and explore the experiences of Nigerian immigrants, particularly in the context of transnational intergenerational caregiving. The paucity of literature on elder care within diaspora communities reflects a critical gap in understanding how caregiving responsibilities are negotiated across distance, culture, and institutional systems. Addressing this gap requires innovative methodologies, including longitudinal tracking of care exchanges, digital ethnography of transnational support networks, and participatory research with diaspora caregivers and their families. Such approaches can illuminate how care is coordinated, sustained, and emotionally managed across borders, informing more responsive and culturally grounded interventions for this growing population.

## Figures and Tables

**Figure 1 ijerph-23-00002-f001:**
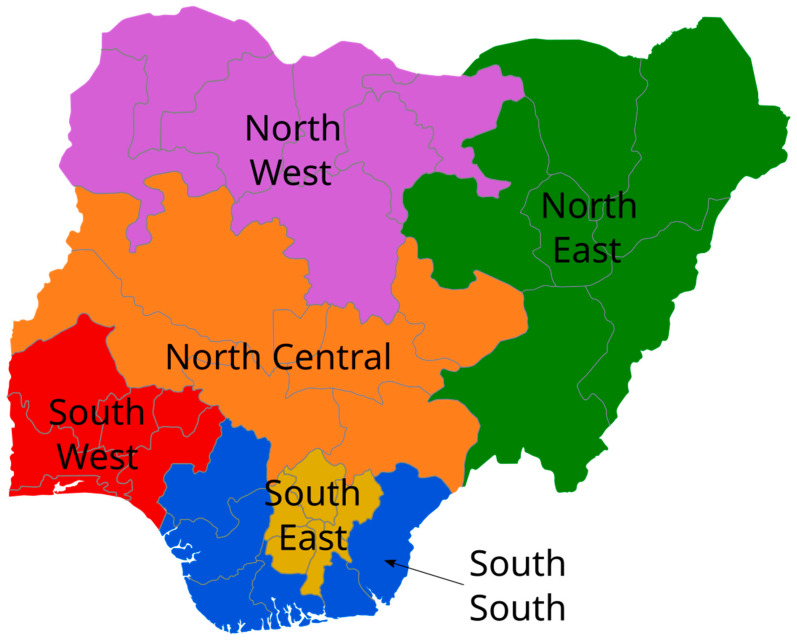
Map of Nigeria showing the six geopolitical zones, the 36 constituent states, and the Federal Capital Territory (FCT), Abuja. Source: Wikimedia Commons. Public domain. https://commons.wikimedia.org/wiki/File:Geopolitical_Zones_of_Nigeria.svg (accessed on 30 October 2025) [4].

**Table 1 ijerph-23-00002-t001:** Overview of Findings.

Themes	Sub-Themes	Analytical Insights
Cultural Influences	High value on respecting and honoring elders Moral duty of younger family members to provide care and support. Filial piety and children’s duty to care for and honor parents	Reflects the persistence of collectivist norms and intergenerational moral obligations shaping caregiving expectations.
Gender Differences	Women as primary caregivers in many households Female inclination to provide caregiving assistance. Challenges faced by adult daughters due to education and financial limitations	Reinforces gendered caregiving roles; highlights need for gender-sensitive policy.
Family Dynamics	Primary responsibility for caregiving lies with the family. Intergenerational care transfer and reliance on younger family members Challenges in extended family system and changing family structures	Indicates strain on traditional systems; suggests need for formal care alternatives.
Economic Factors	Financial limitations and scarce resources Coping strategies of elderly individuals to supplement support. Impact of migration, westernization, changing family structure, urbanization on care and support services	Economic pressures reshape caregiving; urbanization may erode extended family support.
Caregiver Strain and Psychosocial Dimensions	Limited access to healthcare services Balancing multiple responsibilities demands Emotional and physical toll on caregivers	Points to systemic gaps; underscores urgency for caregiver support programs.
Government Policies and Support	Targeted social policies. Support care services for elderly individuals	Reveals limited institutional frameworks for elder care, leaving families as the main support system.

## Data Availability

Not applicable.

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
