# Peer review of "Patterns of Elder Caregiving Among Nigerians: An Integrative Review"

_ijerph, 2025, doi:10.3390/ijerph23010002_

Round 1

Reviewer 1 Report

Comments and Suggestions for Authors

This integrative review provides a timely and valuable synthesis of elder caregiving patterns in Nigeria, a topic of growing importance given the country’s rapidly aging population and shifting sociocultural landscape. The use of an intersectional lens and adherence to PRISMA 2020 guidelines strengthen the methodological rigor and relevance of the findings. The six identified themes—cultural influences, gender differences, family dynamics, economic factors, caregiver challenges, and government policy—are well-supported by the literature and coherently presented.

That said, several areas would benefit from revision to enhance clarity, methodological transparency, and scholarly impact:

Methodological Limitations: The reliance on single-reviewer screening and data extraction increases the risk of selection and interpretation bias. While acknowledged in the limitations, this concern should be more prominently addressed in the Methods section (e.g., by describing steps taken to mitigate subjectivity, such as consultation with a librarian or pilot testing of the extraction matrix). Consider discussing how this limitation may have influenced thematic development.

Geopolitical Representation: The manuscript states that studies “encompassed all six geopolitical zones,” yet the data extraction table (Appendix A) shows uneven coverage (e.g., no studies from North-East or North-West zones are listed). Please clarify this discrepancy—either by confirming representation across all zones or revising the claim to reflect actual regional coverage.

Redundancy and Flow: Some sections, particularly in the Discussion (e.g., paragraphs on filial piety and migration), repeat points already made in the Results. Streamlining the Discussion to focus on synthesis, theoretical implications (e.g., how intersectionality manifests in the Nigerian context), and novel insights—rather than reiterating findings—would improve readability and analytical depth.

Policy Recommendations: The recommendations (Section 4.2) are practical but somewhat generic. Strengthen them by linking each point directly to specific findings from the reviewed studies (e.g., “Given that 60% of caregivers are women facing financial constraints [Namadi, 2016], policy #3 should include targeted income support for female caregivers”).

Transnational Care Gap: The review rightly notes the scarcity of literature on transnational elder caregiving among Nigerian diasporas. Consider expanding this point in the Conclusion to highlight not only the need for future research but also potential methodological approaches (e.g., comparative or mixed-methods studies involving both Nigeria and host countries).

Registration and Protocol: The review was not registered and no protocol was prepared. While not mandatory for integrative reviews, registration (e.g., in PROSPERO or OSF) enhances reproducibility. For future work, consider preregistering review protocols.

Overall, this is a strong and socially relevant review that makes a meaningful contribution to gerontology and global health literature. With the suggested revisions, it will be even more robust and impactful.

Comments on the Quality of English Language

Minor Language Edits: While the English is generally clear and academic, a few sentences are wordy or contain minor grammatical inconsistencies (e.g., “The study highlights the intricate nature…” appears twice in the Abstract and Conclusion). A final proofread would polish the manuscript.

Author Response

We are grateful for your thorough review and valuable comments on our manuscript. Your observations and suggestions have been instrumental in improving the quality and clarity of our work. Below, we provide detailed responses to each point raised and outline the changes that have been made to the manuscript. All changes are highlighted in red in the revised document. We hope that the revisions meet your expectations and improve the rigour and clarity of the paper.

ID

Comment (verbatim)

Response

R1-01

Methodological Limitations: The reliance on single-reviewer screening and data extraction increases the risk of selection and interpretation bias. While acknowledged in the limitations, this concern should be more prominently addressed in the Methods section (e.g., by describing steps taken to mitigate subjectivity, such as consultation with a librarian or pilot testing of the extraction matrix). Consider discussing how this limitation may have influenced thematic development.

Thank you for highlighting this. We revised the Methods section as suggested.

R1-02

Geopolitical Representation: The manuscript states that studies “encompassed all six geopolitical zones,” yet the data extraction table (Appendix A) shows uneven coverage (e.g., no studies from North-East or North-West zones are listed). Please clarify this discrepancy—either by confirming representation across all zones or revising the claim to reflect actual regional coverage.

Thank you for noting this. The statement was revised to reflect actual regional coverage

R1-03

Redundancy and Flow: Some sections, particularly in the Discussion (e.g., paragraphs on filial piety and migration), repeat points already made in the Results. Streamlining the Discussion to focus on synthesis, theoretical implications (e.g., how intersectionality manifests in the Nigerian context), and novel insights—rather than reiterating findings—would improve readability and analytical depth.

Thank you for the feedback. The Discussion has been streamlined.

R1-04

Policy Recommendations: The recommendations (Section 4.2) are practical but somewhat generic. Strengthen them by linking each point directly to specific findings from the reviewed studies (e.g., “Given that 60% of caregivers are women facing financial constraints [Namadi, 2016], policy #3 should include targeted income support for female caregivers”).

Section 4.2 has been revised to link each recommendation to specific findings

R1-05

Transnational Care Gap: The review rightly notes the scarcity of literature on transnational elder caregiving among Nigerian diasporas. Consider expanding this point in the Conclusion to highlight not only the need for future research but also potential methodological approaches (e.g., comparative or mixed-methods studies involving both Nigeria and host countries).

As suggested, we have refined the conclusion to highlight the transnational care gap and propose further methodological directions.

R1-06

Registration and Protocol: The review was not registered and no protocol was prepared. While not mandatory for integrative reviews, registration (e.g., in PROSPERO or OSF) enhances reproducibility. For future work, consider preregistering review protocols.

Thank you for this helpful suggestion.

R2-01.a

The introduction section provides a lot of information about the development of aging, and it would be better to be more concise.

Thank you for the helpful guidance. We have made the introduction more consice

R2-01.b

The necessity of the study and the existing knowledge gap are not stated.

Thank you for noting this. The Introduction was revised to clearly state the study’s necessity and identify the existing knowledge gap.

R2-01.c

The purpose of the study should be clearly stated at the end of the introduction, and the innovation of the study should be stated.

Thank you. This has now been done.

R2-02.a

Lack of Protocol Registration: The review was not registered (e.g., in PROSPERO), and no prior protocol was prepared.

Thank you for noting this. We acknowledge this in the paper.

R2-02.b

Single Reviewer Process: Both screening and data extraction were conducted by a single reviewer without independent verification. This introduces a significant risk of selection bias and data extraction errors. The absence of inter-rater reliability or validation severely weakens methodological robustness.

We recognize this imitation and acknowledge this transparently in the manuscript.

R2-02.c

Although PRISMA 2020 is mentioned, several key elements are missing — including detailed search strategy appendices, full inclusion/exclusion justification, and clear reporting of bias assessment results.

We have moved PRISMA to a supplementary file as we feel it may be of use or some readers to show transparency in the steps that were undertaken in this review.

R2-02.d

Absence of Formal Quality Grading: The authors mention SANRA and MMAT, but results are summarized narratively without clear quantitative presentation. The lack of a table summarizing the quality scores or criteria for classification (high/medium/low) reduces transparency.

The SANRA and MMAT, formerly in Appendices B to E, are now presented as supplementary tables with clear location guidance. Their application in evaluating study quality is detailed in the Methods section.

R2-02.e

No Assessment of Publication Bias or Certainty of Evidence:

As this research area progresses and more studies become available, these assessments will increase in importance.  We acknowledge this as a limitation in the current study.

R2-02.f

Despite acknowledging the absence of bias evaluation, no effort was made to assess missing studies or reporting bias. Given the limited scope (n=20 studies), this omission is critical.

Thank you for noting this. A formal bias assessment was not conducted, as we prioritized conceptual interpretation over quantitative evaluation.  As this field of research continues to develop from a very limited and emerging scope, we agree with the reviewer that future studies should include this.  Therefore, we believe that our acknowledgement of this bias although it not ideal, is fair in the context of the state of evidence in this area.

R2-03.a

While the paper claims to use intersectionality as a guiding framework, it does not apply this lens consistently throughout the analysis.

Thank you for noting this. Careful revisions were made to more consistently apply the framework.

R2-03.b

The discussion fails to connect intersecting factors (gender, socioeconomic status, migration) in a theoretically integrated way.

Thank you for the feedback. The Discussion has been revised to more carefully integrate gender, class, and migration within an intersectional framework.

R2-03.c

The theoretical framework is presented in detail but not operationalized in the results or discussion. Themes are descriptive rather than analytical, limiting theoretical contribution.

Thank you for noting this. The framework is now explicitly linked to the themes

R2-03.d

Thematic findings mostly summarize existing studies without critical comparison or identification of contradictions.

Thank you for noting this. The analysis was revised to include comparative insights and highlight areas of divergence across studies.

R2-03.e

There is little synthesis beyond listing patterns. The analysis remains surface-level and repetitive across sections (e.g., family dynamics, cultural influence, and gender overlap heavily).

Thank you for the observation. Revisions were made to enhance synthesis, reduce overlap, and enhance analytical depth across related sections.

R2-03.f

There is no clear conceptual model or framework emerging from the synthesis.

A named framework now structures the synthesis and guides recommendations.

R2-03.g

Studies of different methodological rigor are treated equally in the synthesis. This dilutes the credibility of conclusions.

We agree that they are treated equally in the sysnthesis. We note this in the limitations section and encourage future studies to address this point.

R2-03.h

Findings are not adequately contextualized within broader African or global caregiving literature.

Thanks for the insight. The Discussion now places the findings within wider African caregiving contexts.

R2-03.i

The paper misses an opportunity to compare Nigerian caregiving trends to those in similar socio-economic contexts.

Thank you for the feedback. The Discussion now compares Nigerian and similar African caregiving contexts.

R2-03.j

The manuscript is unnecessarily long, with repetitive descriptions of cultural norms and caregiving practices. Condensing and synthesizing findings would improve readability and focus.

Thank you for the observation. The manuscript was condensed to remove repetition and streamline descriptions for clearer focus and improved readability.

R2-03.k

There are inconsistencies in citation style (some APA-like, others numeric) and redundancy in referencing.

All citations have been standardized to APA style, and redundant references have been removed for clarity and consistency.

R2-03.l

Several statements are not directly linked to supporting sources.

All key statements are now supported with appropriate citations, ensuring alignment between claims and referenced sources.

R2-03.m

Table 2 and Figure 2 are descriptive but add limited analytical value.

Thank you for the feedback. We revised the table to include an analytical column. The PRISMA flow diagram has been moved to a supplemental file.

R2-03.n

The PRISMA flow diagram lacks detail on the reasons for exclusion at each stage.

Thank you for the comment. The PRISMA flow diagram follows the 2020 guideline, and reasons for exclusion are summarized in the Methods section to avoid redundancy. The PRISMA flow diagram has been moved to a supplemental file (refer to R2-02.c) .

R2-03.o

Although limitations are briefly listed (single reviewer, exclusion of grey literature, etc.), there is no discussion of how these limitations might have influenced results or interpretations.

Thank you. The limitations section has been expanded to explain how these factors may have influenced the findings and interpretations.

R2-03.p

The authors could have discussed the potential for selection bias, linguistic bias (English-only studies), and the lack of local Nigerian-language studies

Thank you. This has been addressed by noting selection and linguistic bias and the exclusion of local-language studies.

Reviewer 2 Report

Comments and Suggestions for Authors

Dear author, I am reviewing your article intitled: "Patterns of Elder Caregiving Among Nigerians: Integrative Review ". This article has strengths and weaknesses, and I hope that my comments will be effective in improving your work.

  • The introduction section provides a lot of information about the development of aging, and it would be better to be more concise.
  • The necessity of the study and the existing knowledge gap are not stated.
  • The purpose of the study should be clearly stated at the end of the introduction, and the innovation of the study should be stated.
  • Among the important weaknesses that can be pointed out in the working method are the following:
  • Lack of Protocol Registration: The review was not registered (e.g., in PROSPERO), and no prior protocol was prepared.
  • Single Reviewer Process: Both screening and data extraction were conducted by a single reviewer without independent verification. This introduces a significant risk of selection bias and data extraction errors. The absence of inter-rater reliability or validation severely weakens methodological robustness.
  • Although PRISMA 2020 is mentioned, several key elements are missing — including detailed search strategy appendices, full inclusion/exclusion justification, and clear reporting of bias assessment results.
  • Absence of Formal Quality Grading: The authors mention SANRA and MMAT, but results are summarized narratively without clear quantitative presentation. The lack of a table summarizing the quality scores or criteria for classification (high/medium/low) reduces transparency.
  • No Assessment of Publication Bias or Certainty of Evidence:

Despite acknowledging the absence of bias evaluation, no effort was made to assess missing studies or reporting bias. Given the limited scope (n=20 studies), this omission is critical.

  • While the paper claims to use intersectionality as a guiding framework, it does not apply this lens consistently throughout the analysis. The discussion fails to connect intersecting factors (gender, socioeconomic status, migration) in a theoretically integrated way.
  • The theoretical framework is presented in detail but not operationalized in the results or discussion. Themes are descriptive rather than analytical, limiting theoretical contribution.
  • Thematic findings mostly summarize existing studies without critical comparison or identification of contradictions. There is little synthesis beyond listing patterns.
  • The analysis remains surface-level and repetitive across sections (e.g., family dynamics, cultural influence, and gender overlap heavily). There is no clear conceptual model or framework emerging from the synthesis.
  • Studies of different methodological rigor are treated equally in the synthesis. This dilutes the credibility of conclusions.
  • Findings are not adequately contextualized within broader African or global caregiving literature. The paper misses an opportunity to compare Nigerian caregiving trends to those in similar socio-economic contexts.
  • The manuscript is unnecessarily long, with repetitive descriptions of cultural norms and caregiving practices. Condensing and synthesizing findings would improve readability and focus.
  • There are inconsistencies in citation style (some APA-like, others numeric) and redundancy in referencing. Several statements are not directly linked to supporting sources.
  • Table 2 and Figure 2 are descriptive but add limited analytical value. The PRISMA flow diagram lacks detail on the reasons for exclusion at each stage.
  • Although limitations are briefly listed (single reviewer, exclusion of grey literature, etc.), there is no discussion of how these limitations might have influenced results or interpretations. The authors could have discussed the potential for selection bias, linguistic bias (English-only studies), and the lack of local Nigerian-language studies

Author Response

We are grateful for your thorough review and valuable comments on our manuscript. Your observations and suggestions have been instrumental in improving the quality and clarity of our work. Below, we provide detailed responses to each point raised and outline the changes that have been made to the manuscript. All changes are highlighted in red in the revised document. We hope that the revisions meet your expectations and improve the rigour and clarity of the paper.

ID

Comment (verbatim)

Response

R1-01

Methodological Limitations: The reliance on single-reviewer screening and data extraction increases the risk of selection and interpretation bias. While acknowledged in the limitations, this concern should be more prominently addressed in the Methods section (e.g., by describing steps taken to mitigate subjectivity, such as consultation with a librarian or pilot testing of the extraction matrix). Consider discussing how this limitation may have influenced thematic development.

Thank you for highlighting this. We revised the Methods section as suggested.

R1-02

Geopolitical Representation: The manuscript states that studies “encompassed all six geopolitical zones,” yet the data extraction table (Appendix A) shows uneven coverage (e.g., no studies from North-East or North-West zones are listed). Please clarify this discrepancy—either by confirming representation across all zones or revising the claim to reflect actual regional coverage.

Thank you for noting this. The statement was revised to reflect actual regional coverage

R1-03

Redundancy and Flow: Some sections, particularly in the Discussion (e.g., paragraphs on filial piety and migration), repeat points already made in the Results. Streamlining the Discussion to focus on synthesis, theoretical implications (e.g., how intersectionality manifests in the Nigerian context), and novel insights—rather than reiterating findings—would improve readability and analytical depth.

Thank you for the feedback. The Discussion has been streamlined.

R1-04

Policy Recommendations: The recommendations (Section 4.2) are practical but somewhat generic. Strengthen them by linking each point directly to specific findings from the reviewed studies (e.g., “Given that 60% of caregivers are women facing financial constraints [Namadi, 2016], policy #3 should include targeted income support for female caregivers”).

Section 4.2 has been revised to link each recommendation to specific findings

R1-05

Transnational Care Gap: The review rightly notes the scarcity of literature on transnational elder caregiving among Nigerian diasporas. Consider expanding this point in the Conclusion to highlight not only the need for future research but also potential methodological approaches (e.g., comparative or mixed-methods studies involving both Nigeria and host countries).

As suggested, we have refined the conclusion to highlight the transnational care gap and propose further methodological directions.

R1-06

Registration and Protocol: The review was not registered and no protocol was prepared. While not mandatory for integrative reviews, registration (e.g., in PROSPERO or OSF) enhances reproducibility. For future work, consider preregistering review protocols.

Thank you for this helpful suggestion.

R2-01.a

The introduction section provides a lot of information about the development of aging, and it would be better to be more concise.

Thank you for the helpful guidance. We have made the introduction more consice

R2-01.b

The necessity of the study and the existing knowledge gap are not stated.

Thank you for noting this. The Introduction was revised to clearly state the study’s necessity and identify the existing knowledge gap.

R2-01.c

The purpose of the study should be clearly stated at the end of the introduction, and the innovation of the study should be stated.

Thank you. This has now been done.

R2-02.a

Lack of Protocol Registration: The review was not registered (e.g., in PROSPERO), and no prior protocol was prepared.

Thank you for noting this. We acknowledge this in the paper.

R2-02.b

Single Reviewer Process: Both screening and data extraction were conducted by a single reviewer without independent verification. This introduces a significant risk of selection bias and data extraction errors. The absence of inter-rater reliability or validation severely weakens methodological robustness.

We recognize this imitation and acknowledge this transparently in the manuscript.

R2-02.c

Although PRISMA 2020 is mentioned, several key elements are missing — including detailed search strategy appendices, full inclusion/exclusion justification, and clear reporting of bias assessment results.

We have moved PRISMA to a supplementary file as we feel it may be of use or some readers to show transparency in the steps that were undertaken in this review.

R2-02.d

Absence of Formal Quality Grading: The authors mention SANRA and MMAT, but results are summarized narratively without clear quantitative presentation. The lack of a table summarizing the quality scores or criteria for classification (high/medium/low) reduces transparency.

The SANRA and MMAT, formerly in Appendices B to E, are now presented as supplementary tables with clear location guidance. Their application in evaluating study quality is detailed in the Methods section.

R2-02.e

No Assessment of Publication Bias or Certainty of Evidence:

As this research area progresses and more studies become available, these assessments will increase in importance.  We acknowledge this as a limitation in the current study.

R2-02.f

Despite acknowledging the absence of bias evaluation, no effort was made to assess missing studies or reporting bias. Given the limited scope (n=20 studies), this omission is critical.

Thank you for noting this. A formal bias assessment was not conducted, as we prioritized conceptual interpretation over quantitative evaluation.  As this field of research continues to develop from a very limited and emerging scope, we agree with the reviewer that future studies should include this.  Therefore, we believe that our acknowledgement of this bias although it not ideal, is fair in the context of the state of evidence in this area.

R2-03.a

While the paper claims to use intersectionality as a guiding framework, it does not apply this lens consistently throughout the analysis.

Thank you for noting this. Careful revisions were made to more consistently apply the framework.

R2-03.b

The discussion fails to connect intersecting factors (gender, socioeconomic status, migration) in a theoretically integrated way.

Thank you for the feedback. The Discussion has been revised to more carefully integrate gender, class, and migration within an intersectional framework.

R2-03.c

The theoretical framework is presented in detail but not operationalized in the results or discussion. Themes are descriptive rather than analytical, limiting theoretical contribution.

Thank you for noting this. The framework is now explicitly linked to the themes

R2-03.d

Thematic findings mostly summarize existing studies without critical comparison or identification of contradictions.

Thank you for noting this. The analysis was revised to include comparative insights and highlight areas of divergence across studies.

R2-03.e

There is little synthesis beyond listing patterns. The analysis remains surface-level and repetitive across sections (e.g., family dynamics, cultural influence, and gender overlap heavily).

Thank you for the observation. Revisions were made to enhance synthesis, reduce overlap, and enhance analytical depth across related sections.

R2-03.f

There is no clear conceptual model or framework emerging from the synthesis.

A named framework now structures the synthesis and guides recommendations.

R2-03.g

Studies of different methodological rigor are treated equally in the synthesis. This dilutes the credibility of conclusions.

We agree that they are treated equally in the sysnthesis. We note this in the limitations section and encourage future studies to address this point.

R2-03.h

Findings are not adequately contextualized within broader African or global caregiving literature.

Thanks for the insight. The Discussion now places the findings within wider African caregiving contexts.

R2-03.i

The paper misses an opportunity to compare Nigerian caregiving trends to those in similar socio-economic contexts.

Thank you for the feedback. The Discussion now compares Nigerian and similar African caregiving contexts.

R2-03.j

The manuscript is unnecessarily long, with repetitive descriptions of cultural norms and caregiving practices. Condensing and synthesizing findings would improve readability and focus.

Thank you for the observation. The manuscript was condensed to remove repetition and streamline descriptions for clearer focus and improved readability.

R2-03.k

There are inconsistencies in citation style (some APA-like, others numeric) and redundancy in referencing.

All citations have been standardized to APA style, and redundant references have been removed for clarity and consistency.

R2-03.l

Several statements are not directly linked to supporting sources.

All key statements are now supported with appropriate citations, ensuring alignment between claims and referenced sources.

R2-03.m

Table 2 and Figure 2 are descriptive but add limited analytical value.

Thank you for the feedback. We revised the table to include an analytical column. The PRISMA flow diagram has been moved to a supplemental file.

R2-03.n

The PRISMA flow diagram lacks detail on the reasons for exclusion at each stage.

Thank you for the comment. The PRISMA flow diagram follows the 2020 guideline, and reasons for exclusion are summarized in the Methods section to avoid redundancy. The PRISMA flow diagram has been moved to a supplemental file (refer to R2-02.c) .

R2-03.o

Although limitations are briefly listed (single reviewer, exclusion of grey literature, etc.), there is no discussion of how these limitations might have influenced results or interpretations.

Thank you. The limitations section has been expanded to explain how these factors may have influenced the findings and interpretations.

R2-03.p

The authors could have discussed the potential for selection bias, linguistic bias (English-only studies), and the lack of local Nigerian-language studies

Thank you. This has been addressed by noting selection and linguistic bias and the exclusion of local-language studies.

We are grateful for your thorough review and valuable comments on our manuscript. Your observations and suggestions have been instrumental in improving the quality and clarity of our work. Below, we provide detailed responses to each point raised and outline the changes that have been made to the manuscript. All changes are highlighted in red in the revised document. We hope that the revisions meet your expectations and improve the rigour and clarity of the paper.

ID

Comment (verbatim)

Response

R1-01

Methodological Limitations: The reliance on single-reviewer screening and data extraction increases the risk of selection and interpretation bias. While acknowledged in the limitations, this concern should be more prominently addressed in the Methods section (e.g., by describing steps taken to mitigate subjectivity, such as consultation with a librarian or pilot testing of the extraction matrix). Consider discussing how this limitation may have influenced thematic development.

Thank you for highlighting this. We revised the Methods section as suggested.

R1-02

Geopolitical Representation: The manuscript states that studies “encompassed all six geopolitical zones,” yet the data extraction table (Appendix A) shows uneven coverage (e.g., no studies from North-East or North-West zones are listed). Please clarify this discrepancy—either by confirming representation across all zones or revising the claim to reflect actual regional coverage.

Thank you for noting this. The statement was revised to reflect actual regional coverage

R1-03

Redundancy and Flow: Some sections, particularly in the Discussion (e.g., paragraphs on filial piety and migration), repeat points already made in the Results. Streamlining the Discussion to focus on synthesis, theoretical implications (e.g., how intersectionality manifests in the Nigerian context), and novel insights—rather than reiterating findings—would improve readability and analytical depth.

Thank you for the feedback. The Discussion has been streamlined.

R1-04

Policy Recommendations: The recommendations (Section 4.2) are practical but somewhat generic. Strengthen them by linking each point directly to specific findings from the reviewed studies (e.g., “Given that 60% of caregivers are women facing financial constraints [Namadi, 2016], policy #3 should include targeted income support for female caregivers”).

Section 4.2 has been revised to link each recommendation to specific findings

R1-05

Transnational Care Gap: The review rightly notes the scarcity of literature on transnational elder caregiving among Nigerian diasporas. Consider expanding this point in the Conclusion to highlight not only the need for future research but also potential methodological approaches (e.g., comparative or mixed-methods studies involving both Nigeria and host countries).

As suggested, we have refined the conclusion to highlight the transnational care gap and propose further methodological directions.

R1-06

Registration and Protocol: The review was not registered and no protocol was prepared. While not mandatory for integrative reviews, registration (e.g., in PROSPERO or OSF) enhances reproducibility. For future work, consider preregistering review protocols.

Thank you for this helpful suggestion.

R2-01.a

The introduction section provides a lot of information about the development of aging, and it would be better to be more concise.

Thank you for the helpful guidance. We have made the introduction more consice

R2-01.b

The necessity of the study and the existing knowledge gap are not stated.

Thank you for noting this. The Introduction was revised to clearly state the study’s necessity and identify the existing knowledge gap.

R2-01.c

The purpose of the study should be clearly stated at the end of the introduction, and the innovation of the study should be stated.

Thank you. This has now been done.

R2-02.a

Lack of Protocol Registration: The review was not registered (e.g., in PROSPERO), and no prior protocol was prepared.

Thank you for noting this. We acknowledge this in the paper.

R2-02.b

Single Reviewer Process: Both screening and data extraction were conducted by a single reviewer without independent verification. This introduces a significant risk of selection bias and data extraction errors. The absence of inter-rater reliability or validation severely weakens methodological robustness.

We recognize this imitation and acknowledge this transparently in the manuscript.

R2-02.c

Although PRISMA 2020 is mentioned, several key elements are missing — including detailed search strategy appendices, full inclusion/exclusion justification, and clear reporting of bias assessment results.

We have moved PRISMA to a supplementary file as we feel it may be of use or some readers to show transparency in the steps that were undertaken in this review.

R2-02.d

Absence of Formal Quality Grading: The authors mention SANRA and MMAT, but results are summarized narratively without clear quantitative presentation. The lack of a table summarizing the quality scores or criteria for classification (high/medium/low) reduces transparency.

The SANRA and MMAT, formerly in Appendices B to E, are now presented as supplementary tables with clear location guidance. Their application in evaluating study quality is detailed in the Methods section.

R2-02.e

No Assessment of Publication Bias or Certainty of Evidence:

As this research area progresses and more studies become available, these assessments will increase in importance.  We acknowledge this as a limitation in the current study.

R2-02.f

Despite acknowledging the absence of bias evaluation, no effort was made to assess missing studies or reporting bias. Given the limited scope (n=20 studies), this omission is critical.

Thank you for noting this. A formal bias assessment was not conducted, as we prioritized conceptual interpretation over quantitative evaluation.  As this field of research continues to develop from a very limited and emerging scope, we agree with the reviewer that future studies should include this.  Therefore, we believe that our acknowledgement of this bias although it not ideal, is fair in the context of the state of evidence in this area.

R2-03.a

While the paper claims to use intersectionality as a guiding framework, it does not apply this lens consistently throughout the analysis.

Thank you for noting this. Careful revisions were made to more consistently apply the framework.

R2-03.b

The discussion fails to connect intersecting factors (gender, socioeconomic status, migration) in a theoretically integrated way.

Thank you for the feedback. The Discussion has been revised to more carefully integrate gender, class, and migration within an intersectional framework.

R2-03.c

The theoretical framework is presented in detail but not operationalized in the results or discussion. Themes are descriptive rather than analytical, limiting theoretical contribution.

Thank you for noting this. The framework is now explicitly linked to the themes

R2-03.d

Thematic findings mostly summarize existing studies without critical comparison or identification of contradictions.

Thank you for noting this. The analysis was revised to include comparative insights and highlight areas of divergence across studies.

R2-03.e

There is little synthesis beyond listing patterns. The analysis remains surface-level and repetitive across sections (e.g., family dynamics, cultural influence, and gender overlap heavily).

Thank you for the observation. Revisions were made to enhance synthesis, reduce overlap, and enhance analytical depth across related sections.

R2-03.f

There is no clear conceptual model or framework emerging from the synthesis.

A named framework now structures the synthesis and guides recommendations.

R2-03.g

Studies of different methodological rigor are treated equally in the synthesis. This dilutes the credibility of conclusions.

We agree that they are treated equally in the sysnthesis. We note this in the limitations section and encourage future studies to address this point.

R2-03.h

Findings are not adequately contextualized within broader African or global caregiving literature.

Thanks for the insight. The Discussion now places the findings within wider African caregiving contexts.

R2-03.i

The paper misses an opportunity to compare Nigerian caregiving trends to those in similar socio-economic contexts.

Thank you for the feedback. The Discussion now compares Nigerian and similar African caregiving contexts.

R2-03.j

The manuscript is unnecessarily long, with repetitive descriptions of cultural norms and caregiving practices. Condensing and synthesizing findings would improve readability and focus.

Thank you for the observation. The manuscript was condensed to remove repetition and streamline descriptions for clearer focus and improved readability.

R2-03.k

There are inconsistencies in citation style (some APA-like, others numeric) and redundancy in referencing.

All citations have been standardized to APA style, and redundant references have been removed for clarity and consistency.

R2-03.l

Several statements are not directly linked to supporting sources.

All key statements are now supported with appropriate citations, ensuring alignment between claims and referenced sources.

R2-03.m

Table 2 and Figure 2 are descriptive but add limited analytical value.

Thank you for the feedback. We revised the table to include an analytical column. The PRISMA flow diagram has been moved to a supplemental file.

R2-03.n

The PRISMA flow diagram lacks detail on the reasons for exclusion at each stage.

Thank you for the comment. The PRISMA flow diagram follows the 2020 guideline, and reasons for exclusion are summarized in the Methods section to avoid redundancy. The PRISMA flow diagram has been moved to a supplemental file (refer to R2-02.c) .

R2-03.o

Although limitations are briefly listed (single reviewer, exclusion of grey literature, etc.), there is no discussion of how these limitations might have influenced results or interpretations.

Thank you. The limitations section has been expanded to explain how these factors may have influenced the findings and interpretations.

R2-03.p

The authors could have discussed the potential for selection bias, linguistic bias (English-only studies), and the lack of local Nigerian-language studies

Thank you. This has been addressed by noting selection and linguistic bias and the exclusion of local-language studies.